# High-Dispersed V_2_O_5_-CuO_X_ Nanoparticles on h-BN in NH_3_-SCR and NH_3_-SCO Performance

**DOI:** 10.3390/nano12142329

**Published:** 2022-07-06

**Authors:** Han-Gyu Im, Myeung-Jin Lee, Woon-Gi Kim, Su-Jin Kim, Bora Jeong, Bora Ye, Heesoo Lee, Hong-Dae Kim

**Affiliations:** 1Industrial Environment Green Deal Agency, Korea Institute of Industrial Technology, Ulsan 44413, Korea; hangyu@kitech.re.kr (H.-G.I.); leemj@kitech.re.kr (M.-J.L.); woongi25@kitech.re.kr (W.-G.K.); sujini@kitech.re.kr (S.-J.K.); bora1106@kitech.re.kr (B.J.); yebora@kitech.re.kr (B.Y.); 2Department of Materials Science & Engineering, Pusan National University, Busan 46241, Korea

**Keywords:** V-Cu-based catalyst, NH_3_-Slip, hexagonal boron nitride, selective catalytic reduction, selective catalytic oxidation

## Abstract

Typically, to meet emission regulations, the selective catalytic reduction of NO_X_ with NH_3_ (NH_3_-SCR) technology cause NH_3_ emissions owing to high NH_3_/NO_X_ ratios to meet emission regulations. In this study, V-Cu/BN-Ti was used to remove residual NO_X_ and NH_3_. Catalysts were evaluated for selective catalytic oxidation of NH_3_ (NH_3_-SCO) in the NH_3_-SCR reaction at 200–300 °C. The addition of vanadium and copper increased the number of Brønsted and Lewis acid sites available for the reaction by increasing the ratio of V^5+^ and forming Cu^+^ species, respectively. Furthermore, h-BN was dispersed in the catalyst to improve the content of vanadium and copper species on the surface. NH_3_ and NO_X_ conversion were 98% and 91% at 260 °C, respectively. Consequently, slipped NH_3_ (NH_3_-Slip) emitted only 2% of the injected ammonia. Under SO_2_ conditions, based on the NH_3_ oxidation reaction, catalytic deactivation was improved by addition of h-BN. This study suggests that h-BN is a potential catalyst that can help remove residual NO_X_ and meet NH_3_ emission regulations when placed at the bottom of the SCR catalyst layer in coal-fired power plants.

## 1. Introduction

Owing to the developments in industries and the accompanying increase in fuel consumption, air pollution has increased significantly. In particular, NO_X_, one of the primary air pollutants, is harmful to the human body itself [1]. NO_X_ is a compound of nitrogen and oxygen, including NO, NO_2_, N_2_O, and N_2_O_3_; it can result in ozone layer depletion, greenhouse effect, photochemical smog, and acid rain. Accordingly, various environmental protection regulations have been strengthened, and NO_X_ emission standards have become more stringent [2,3]. Generally, most NO_X_ emissions originate from combustion in stationary, such as coal-fired power plants [4,5]. Among the existing control techniques of NO_X_, the selective catalyst reduction of NO_X_ with NH_3_ (NH_3_-SCR), which entails the use of ammonia as a reductant, is one of the most commonly applied techniques in stationary likes coal-fired power plant owing to its convenient operation and maintenance and effective NO_X_ conversion performance [6]. The commercial performance of NH_3_-SCR can reach 90% when using V_2_O_5_-WO_3_/TiO_2_ as the catalyst at 300–450 °C [7,8,9]. The corresponding working principle can be expressed as follows:(1)$$4\mathrm{NO}+4{\mathrm{NH}}_{3}+O_{2}\to 4N_{2}+6H_{2}O$$
(2)$$2{\mathrm{NO}}_{2}+4{\mathrm{NH}}_{3}+O_{2}\to 3N_{2}+6H_{2}O$$
(3)$$6\mathrm{NO}+4{\mathrm{NH}}_{3}\to 5N_{2}+6H_{2}O$$
(4)$$6{\mathrm{NO}}_{2}+8{\mathrm{NH}}_{3}\to 7N_{2}+12H_{2}O$$

However, owing to the significantly high flow rates and non-uniform mixture gases in actual plants, catalytic deactivation and NO_X_ removal efficiency standards may not be met, and residual NO_X_ can be emitted. Therefore, additional ammonia injection is necessary to meet the NO_X_ removal efficiency and NOx emission regulations. However, SO_2_ in flue gas can be easily oxidized to SO_3_ when using a catalyst, which then reacts with the slipped ammonia (NH_3_-Slip) to form sticky ammonium bisulfate (ABS, NH_4_HSO_4_) [10]. Thermal decomposition temperatures of ABS are in the range 300–400 °C. However, when the gas stream reaches the bottom of the catalyst layer, the temperature is sufficiently lower (i.e., <300 °C) to cause continuous ABS formation. The formed ABS covers the catalytic reaction surface, resulting in catalyst deactivation and the rusting of equipment. NH_3_ is a harmful air pollutant because it is a toxic and corrosive gas [11]. Therefore, NH_3_-Slip must be carefully managed by reducing NH_3_ contamination to ensure the stable operation of the SCR system and guarantee the longevity of the catalyst.

Therefore, to reduce NH_3_-Slip, techniques such as catalytic oxidation, combustion, absorption, and adsorption have been developed [12,13,14]. Among these, the selective catalytic oxidation of ammonia (NH_3_-SCO) is an environment-friendly process, because NH_3_ is converted to N_2_ and H_2_O [15]. Therefore, it shows significant potential for the mitigation of NH_3_-Slip. However, overcoming the problem of ammonia emissions from coal-fired power plants remains difficult owing to the high investment costs required; hence, the applicability of the NH_3_-SCO is limited [16]. The oxidation reaction of ammonia can be expressed as follows:(5)$$4{\mathrm{NH}}_{3}+3O_{2}\to 2N_{2}+6H_{2}O$$
(6)$$4{\mathrm{NH}}_{3}+4O_{2}\to 2N_{2}O+6H_{2}O$$
(7)$$4{\mathrm{NH}}_{3}+5O_{2}\to 2\mathrm{NO}+6H_{2}O$$
(8)$$4{\mathrm{NH}}_{3}+7O_{2}\to 2{\mathrm{NO}}_{2}+6H_{2}O$$

To date, many catalysts have been studied for the NH_3_-SCO reaction. These catalysts can be generally classified into three types: noble metal-based, zeolite-based, and transition metal oxides-based catalysts. Noble metal-based catalysts (such as those based on Pt, Pd, Ir, Ag, and Au) typically exhibit high oxidative activities at 200–300 °C [17,18,19,20,21]. However, oxidation of NH_3_ causes NO_X_ production, as expressed in Equations (6)–(8), which occurs readily at high temperatures. These catalysts also exhibit low N_2_ selectivity. Moreover, employing noble metal-based catalysts remains challenging owing to their high costs. Zeolite-based catalysts (such as those based on Cu-CHA, Cu-ZSM-5, Fe-ZSM-5, and Fe-MOR) are ion-exchange catalysts and exhibit high N_2_ selectivity in the NH_3_-SCO reaction. However, the zeolite production process is leading to high prices [22]. For this reason, their application is limited in mobile sources, requiring lower volumes rather than stationary sources. Catalysts for relatively higher volume requirements could be developed by exploring supports based on ceramics [23]. In contrast, transition metal oxides (including CuO, Fe_2_O_3_, MnO_4_, V_2_O_5_, and Co_3_O_4_) are abundant and inexpensive and are also considered as alternatives to noble metal catalysts. Commercially available vanadium-based catalysts have been reported to be suitable for the NH_3_-SCR process owing to the presence of V^5+^; however, the oxidation of NH_3_ remains limited [24]. Copper-based catalysts have generally been studied as NH_3_-SCO catalysts with excellent catalytic properties, such as high N_2_ selectivity and relatively low costs [25,26,27,28,29,30]. In particular, NO_X_ conversion efficiencies of 90% at 350 °C have been reported for the Cu/Ti catalysts, which uses TiO_2_ as a support [31]. Therefore, copper species can be utilized to remove both NH_3_ and NO_X_. However, NH_3_-SCR catalysts are generally exposed to high temperatures owing to the constant operation in the catalyst layer. Therefore, by selecting TiO_2_ as a support, NH_3_-SCR catalyst is resistant to SO_2_ present in the exhaust gas and also to high temperatures [32]. Nevertheless, copper-based catalysts still remain vulnerable to high temperatures and sulfur [33].

Hexagonal boron nitride (h-BN) is a sp^2^-hybridized 2D material, comprising an array of six-membered rings of B and N atoms. Notably, h-BN can be synthesized in the shape of a plate owing to its structural properties, and it promotes the dispersion of catalytic species [34]. In addition, it is considered as a potential material in many research fields owing to its excellent properties, such as high thermal stability and conductivity originating from stable bonding and outstanding chemical stability [35]. These advantages render it suitable for long-term operation under high temperatures during NH_3_-SCR, which also involves toxic atmospheres. Furthermore, owing to its high chemical resistance, h-BN improves the poisoning resistance of copper species to SO_2_ and can result in successful NH_3_ oxidation. Despite these advantages, h-BN catalysts for NH_3_-SCR and the corresponding NH_3_ oxidation processes have not been reported. It is expected that developing and applying the h-BN catalyst with NH_3_-SCO performance to the bottom of the SCR catalyst layer can help remove NH_3_-Slip, along with residual NO_X_ in the exhaust gas; this, in turn, would help reduce the maintenance costs of the NH_3_-SCR system.

In this study, vanadium-based catalysts were synthesized via a simple impregnation method by adding copper and h-BN and compared to commercial V/Ti catalysts. In determining the SCR catalytic performance of synthesized catalysts, variables such as gas hourly space velocity (GHSV), catalyst particle size, and reaction pressure were kept constant. The NH_3_-SCR and NH_3_ oxidation performances were evaluated at a specific temperature (200–300 °C), emulating the bottom of the catalyst layer. The improved redox properties and availability of surface acid sites when vanadium and copper were co-precipitated were analyzed using various analytical techniques. The increased content of elements on the surface and improved ratio of V^5+^ and Cu^+^ species increased the number of Brønsted and Lewis acid sites available for SCR and SCO reactions, respectively. Thus, this study demonstrates the effective removal of residual NO_X_ originating from the NH_3_-SCR process in coal-fired power plants.

## 2. Materials and Methods

### 2.1. Catalyst Preparation

All the catalysts used in this study were synthesized via the impregnation method for the selective catalytic oxidation of NH_3_ in the NH_3_-SCR process. First, oxalic acid (HO_2_CCO_2_H, ≥99.0, Sigma-Aldrich, St. Louis, MO, USA) was mixed with 50 mL of ethanol to dissolve the vanadium precursor. To control the oxidation number of vanadium, ammonium metavanadate (NH_4_VO_3_, 99%, Sigma-Aldrich) was mixed with citric acid (HOC(COOH)(CH_2_COOH)_2_, ≥99.5%, Sigma-Aldrich) for 1 h at 60 °C. Copper (II) nitrate trihydrate (CuH_6_N_2_O_9_, 99–104%, Sigma-Aldrich) was used as the copper precursor and mixed in 50 mL ethanol. Each metal precursor was loaded in a certain weight ratio to attain metal contents of 1 wt. % V and 5 wt. % Cu. Titanium dioxide (TiO_2_, >97%, NANO Co., Ltd.) and hexagonal boron nitride (BN, 98%, Sigma-Aldrich, St. Louis, MI, USA) were prepared (TiO_2_:h-BN = 10:1) and dissolved in 100 mL of ethanol and sonicated for 1 h using a sonicator (UP400St, Hielscher, Teltow, Germany) with a 7 mm tip and power of 200 W to ensure uniform dispersion. For impregnation, the vanadium and copper solution was added to the h-BN suspension, stirred for 30 min, and then mixed with the TiO_2_ suspension. The resulting suspension was stirred overnight at 80 °C in an oil bath, until all the ethanol was evaporated. The powder was subsequently calcined at 400 °C for 5 h and finely ground. Finally, the catalyst was prepared successfully; it was labelled as V-Cu/BN-Ti. For comparison, V-Cu/Ti, V/Ti, and Cu/Ti were also synthesized via the same impregnation method, resulting in a total of 4 samples.

### 2.2. Characterization

X-ray diffraction (XRD; Ultima IV, Rigaku, Japan) was performed to confirm the crystal phase of each synthesized catalyst with Cu Kα radiation (*λ* = 1.5406 Å) in the 2θ range 20°–90° with a 1°/min scan rate. To remove the absorbed sample impurities, such as water vapor and organic compounds, degassing pretreatment was performed at 150 °C for 6 h. Subsequently, the samples were subjected to flowing N_2_ gas and nitrogen adsorption–desorption isotherms were measured at −196 °C. The corresponding pore size distribution curves were calculated by the Barrett–Joyner–Halenda (BJH) method using an ASAP 2020 instrument (Micromeritics Instrument Crop, Norcross, GA, USA). The specific surface area, pore volume, and pore diameter of catalysts were calculated by the Brunauer–Emmett–Teller (BET) method. Transmission electron microscopy (TEM) and high-resolution transmission electron microscopy (HR-TEM) were performed on a JEM-2100 (JEOL Ltd., Akishima, Tokyo, Japan) to observe the morphology of vanadium and copper oxides lattice on the h-BN surface. To prepare samples, V-Cu/BN-Ti and V-Cu/Ti were dissolved in ethanol and dispersed by ultrasonication for 10 min. The solution was dropped into the carbon film grid and dried overnight under a vacuum oven at 80 °C. X-ray photoelectron spectroscopy (XPS; K Alpha^+^, Thermo VG Scientific, Waltham, MA, USA) was conducted to determine surface contents and chemical states with an Al Kα radiation source. The binding energies of Cu 2p, V 2p, and O 1s were calibrated using adventitious carbon (C 1s = 284.6 eV). Fourier transform infrared spectroscopy (FT-IR; Vertex 80v, Bruker, Billerica, MA, USA) was performed to investigate chemical bonding in the 4000 to 400 cm^−1^ range. KBr pellets were prepared by pressing together 0.16 g KBr and 0.001 g of the synthesized catalyst. All samples were scanned 256 times. NH_3_ temperature-programmed desorption (NH_3_-TPD) was carried out to analyze acid sites (AutoChem II 2920, Micromeritics Instrument Crop, Norcross, GA, USA). Samples were pretreated in an N_2_ atmosphere at 150 °C for 4 h and NH_3_ was adsorbed using 10% NH_3_/He gas at 150 °C for 1 h. Then, NH_3_ desorption was performed over the samples while increasing the temperature 100–800 °C at a 10 °C/min scan rate. To assess the reduction ability, H_2_ temperature-programmed reduction (H_2_-TPR) was also carried out using the same equipment by passing 10% H_2_/Ar gas over the samples while increasing the temperature from 100 to 800 °C with a 10 °C/min scan rate.

### 2.3. Catalytic Performance Test

The catalytic efficiency of the synthesized catalysts in NO_X_ removal from NH_3_-SCR and NH_3_-SCO was evaluated using a fixed-bed quartz reactor. Catalyst powder was prepared at 0.3 g and placed in a reactor. The reactor temperature was increased to 200 °C for 1 h to eliminate water vapor. The preheater and gas line temperatures were set to 350 and 200 °C, respectively. The gas conditions were as follows: 300 ppm NO_X_, 300 ppm NH_3_, 100 ppm SO_2_ (when used), 5 vol. % O_2_, and balanced N_2_. The total flow rate was 500 mL/min, and thus, the gas was allowed to flow at a GHSV of 60,000 h^−1^. The gas concentrations at inlet and outlet were measured using FT-IR (CX-4000, Gasmet, Vantaa, Finland). When the gas stream in the bypassed line was stable, it was switched to the reactor line to pass through the catalyst. The temperature of performance test was increased from 200 to 300 °C in 20 °C intervals. The NO_X_ conversion was quantified from the inlet and outlet gas contents using Equation (9). The NH_3_ conversion was quantified using Equation (10). The quantity of NH_3_-SCO of NH_3_-Slip was calculated using Equation (11). N_2_ selectivity, i.e., the selective conversion of NO, N_2_O, and NH_3_ to N_2_, was calculated considering all gases using Equation (12).
(9)$${\mathrm{NO}}_{X}\;\mathrm{conversion}\;\left(\%\right)=\frac{{\mathrm{NO}}_{X_{\mathrm{inlet}}}-{\mathrm{NO}}_{X_{\mathrm{outlet}}}}{{\mathrm{NO}}_{X_{\mathrm{inlet}}}}\times 100$$
(10)$${\mathrm{NH}}_{3}\;\mathrm{conversion}\;\left(\%\right)=\frac{{\mathrm{NH}}_{3_{\mathrm{inlet}}}-{\mathrm{NH}}_{3_{\mathrm{outlet}}}}{{\mathrm{NH}}_{3_{\mathrm{inlet}}}}\times 100$$
(11)$${\mathrm{NH}}_{3}\;\mathrm{oxidation}\;\left(\%\right)=\frac{{\mathrm{NH}}_{3_{\mathrm{inlet}}}-\left({\mathrm{NO}}_{X_{\mathrm{inlet}}}-{\mathrm{nNO}}_{X_{\mathrm{outlet}}}\right)-{\mathrm{NH}}_{3_{\mathrm{outlet}}}}{{\mathrm{NH}}_{3\_\mathrm{inlet}}}\times 100$$
(12)$$N_{2}\;\mathrm{selectivity}\;\left(\%\right)=\frac{{\mathrm{NO}}_{X_{\mathrm{inlet}}}+{\mathrm{NH}}_{3_{\mathrm{inlet}}}-{\mathrm{NO}}_{X_{\mathrm{outlet}}}-{\mathrm{NH}}_{3_{\mathrm{outlet}}}-2N_{2}O_{\mathrm{outlet}}}{{\mathrm{NO}}_{X_{\mathrm{inlet}}}+{\mathrm{NH}}_{3_{\mathrm{inlet}}}}\times 100$$

## 3. Results and Discussion

### 3.1. Morphology and Textile Properties Analysis

Morphology and textile properties were analyzed to confirm that the synthesized catalysts had the desired physical properties. XRD analysis was performed to examine the crystal structure. Figure 1d shows the XRD patterns of the synthesized catalysts. It can be seen that all catalysts exhibit diffraction peaks at 2θ = 25.3°, 36.9°, 37.7°, 38.5°, 48.0°, 53.8°, 55.0°, 62.6°, 68.9°, 70.3°, and 75.1°. This corresponded to the characteristic peaks of anatase TiO_2_ (PDF card JCPDS#21–1272) and was indexed as (101), (103), (004), (112), (200), (105), (211), (204), (116), (220), and (215), respectively [36,37,38]. In the case of V-Cu/BN-Ti, a peak at 26.74° (JCPDS#34-0421) was confirmed to correspond to (002) of h-BN and suggested that the synthesis was successful. However, there were no peaks for the active species related to vanadium and copper. In the synthesis process, vanadium and copper were impregnated in low quantities compared to TiO_2_ and h-BN. Therefore, it is expected that vanadium and copper particles were covered by TiO_2_ particles or are highly dispersed, resulting in the absence of peaks of the corresponding crystal phases.

For V-Cu/Ti and V-Cu/BN-Ti, morphology characteristics could be observed in TEM images, allowing the calculation of the lattice distance (Figure 1a and Appendix A). The CuO crystal phase was confirmed to be well-formed with interplanar distances of V-Cu/Ti, V-Cu/BN-Ti corresponding to 0.249 nm of CuO (002) (JCPDF#80–0076) [39]. In addition, the crystallization of anatase TiO_2_ was observed, corresponding to the 0.371nm interplanar distance of TiO_2_ (101) [40]. As shown in Figure 1b and Appendix A, most of the particles consisted of anatase TiO_2_; the average particle size was 9 nm. However, many particles were observed to aggregate, which were all found to be TiO_2_ particles. This implied that many active species (viz., vanadium, and copper) were likely to be aggregated, which affected the catalytic activity. In Figure 1a, the CuO crystal plane corresponding to (002) and the anatase TiO_2_ crystal plane of (101) is observed for V-Cu/BN-Ti. Compared to Appendix A, it could be confirmed that CuO particles were formed on the surface of the h-BN particles in V-Cu/BN-Ti. This indicated that the addition of h-BN enhanced the dispersion of the particles. Hence, h-BN prevented particle agglomeration and contributes to the dispersion of the active species on the particle surface. Table 1 shows the percentage of surface-exposed elemental content. The active species content of V-Cu/BN-Ti was higher than that of V-Cu/Ti (V = 3.07% and Cu = 3.70%). Therefore, the addition of h-BN enhanced the surface dispersion of the active species, resulting in improved surface-exposed vanadium and copper. The crystalline phase of vanadium was not observable in the TEM image. However, it was confirmed that vanadium was present on h-BN using EDS mapping of V-Cu/Ti and V-Cu/BN-Ti (Figure 1c and Appendix A).

Appendix A shows the N_2_ adsorption–desorption isotherms and pore size distribution of the synthesized catalysts. The shapes of the isotherms were generally similar and corresponded to typical type IV curves, which indicated the presence of micropores and mesopores [41]. The sample with added h-BN, V-Cu/BN-Ti, showed H3-type hysteresis loops due to the plate-like characteristic of h-BN [42]. The size of the mesopores were confirmed to be approximately 9.1 nm and 10.0 nm for V-Cu/BN-Ti and V-Cu/Ti, respectively, from the pore size distribution calculated by the BJH method. This result was consistent with the pore size determined from TEM images. Appendix A lists the physical parameters, including specific surface area, pore volume, and pore size of the synthesized catalysts. When elements were added sequentially, a decrease in the specific surface area was observed because the content of TiO_2_ with a relatively large specific surface area was decreased. The TEM (Figure 1a–c) and BET results (Appendix A) corroborate this. V-Cu/BN-Ti showed a decline in physical properties, such as specific surface area due to improved particle aggregation and enhanced surface exposure of V_2_O_5_ and CuO and active species. This suggested that active species exposure should lead to increased reaction at the surface, resulting in increased NO_X_ removal efficiency or NH_3_ oxidation.

### 3.2. Characterization of the NH_3_-SCR and NH_3_-SCO Catalysts

XPS analysis was performed to compare the surface components of the synthesized catalyst and the chemical states of the elements. The deconvoluted V 1p, Cu 2p, and O 1s spectra are shown in Figure 2. Figure 2a shows the Cu 2p spectra, consisting of Cu 2p_1/2_, Cu 2p_3/2_, and two satellite peaks. The Cu 2p_1/2_ and Cu 2p_3/2_ peaks located at 932.3–932.7 eV and 952.1–952.5 eV correspond to Cu^+^ species, whereas the Cu 2p_1/2_ and Cu 2p_3/2_ peaks located at 933.5–933.9 eV and 953.4–953.8 eV correspond to Cu^2+^. Shake-up satellite peaks are located approximately 10 eV higher than the peaks corresponding to Cu 2p_3/2_ [43,44]. It has been reported that the presence of Cu^2+^ could contribute to both NH_3_-SCO and NH_3_-SCR performance and that Cu^+^ contributes to NH_3_-SCO [45].

The Cu 2p spectrum of V-Cu/Ti had peaks at 932.0 and 951.5 eV, while that of V-Cu/BN-Ti exhibits peaks at 932.1 and 952.0 eV. Each peak of Cu 2p_3/2_ was observed at lower binding energies, and the peaks at higher binding energy corresponded to the Cu 2p_1/2_ profile of Cu^+^. This suggested that Cu^+^ species were formed on the sample surface when copper was added to the vanadium-based sample and might contribute to oxidation of NH_3_.

In the V 2p_2/3_ spectra (Figure 2b), vanadium species exist on the catalyst surface as V^5+^ (516.3–517.3 eV), V^4+^ (515.3–516.3 eV), and V^3+^ (514.5–515.5 eV) [46,47]. Among the various oxidation states of vanadium, V^5+^ is known to have excellent redox ability in the NH_3_-SCR reaction [48]. In V/Ti, V^5+^ and V^4+^ corresponded to 517.3 and 515.8, respectively. The ratio of V^5+^/V^4+^ was 0.84%. In the case of the catalyst to which copper was added (V/Ti), the peak corresponding to V^4+^ disappeared and all the peaks corresponded to V^5+^. This indicated that most of the vanadium species of the V-Cu-based catalyst formed V^5+^ = O vanadium oxide species in the process of forming crystalline V_2_O_5_.

As shown in Figure 2c, the O 1s peaks could be deconvoluted into three peaks corresponding to chemisorbed (530.7–532.0 eV) and lattice oxygen (529.4–530.4 eV), which were labelled as O_α_ and O_β_, respectively. It has been reported that a higher O_α_ concentration cause higher performance in NH_3_-SCR and NH_3_-SCO reactions because of better oxidability and mobility [49]. Therefore, this could be one of the reasons that the efficiency of NH_3_-SCR and oxidation of NH_3_ of vanadium species and copper species increases as the O_α_ ratio increased.

O_α_/(O_α_ + O_β_) values were calculated from the deconvoluted XPS spectra and the results are shown in Table 1. The O_α_/(O_α_ + O_β_) ratio significantly increased when copper was added to vanadium. This can be attributed to the change in the valence states of copper and vanadium species. h-BN enhanced the surface exposure of the active species. In the case of V-Cu/Ti, there was no significant difference in the exposure of vanadium and copper species. However, in V-Cu/BN-Ti, the exposure of V and Cu significantly increased to 3.07% and 3.70%, respectively. As shown in Figure 1b and Appendix A, this was due to the improvement of the aggregation behavior via the formation of active species on h-BN particles. As a result, the addition of vanadium and copper led to an increased ratio of O_α_ and V^5+^, which is expected to increase the NH_3_-SCR efficiency. In addition, Cu^+^ peaks appeared, and it is expected that NH_3_-SCO would be improved by 0.12%, corresponding to Cu^+^/(Cu^+^ + Cu^2+^). Ultimately, the addition of h-BN promoted the dispersion of these vanadium and copper species and increased the surface exposed vanadium and copper contents.

FT-IR spectra are shown in Figure 3. Peaks at 3434 cm^−1^ and 1630 cm^−1^ correspond to O–H bonds, –OH groups, and OH stretches [50]. This was attributed to Ti–OH corresponding to TiO_2_, which accounted for most of the synthesized samples. In addition, the catalyst to which h-BN was added exhibited new floating peaks at 804 cm^−1^ and 1405 cm^−1^, corresponding to B–N stretching and B–N bending, respectively [51]. In the sample impregnated with vanadium and copper, a change in the IR band was observed in the range 1127–1058 cm^−1^ (Figure 3b) [52]. This corresponded to M–OH stretching and bending and might indicate the formation of many hydroxyl groups in the chemical structure [53]. Additionally, the 400–900 cm^−1^ (Figure 3c) region presented mostly titanium-related peaks. It has been reported that peaks of 536 and 459 cm^−1^ correspond to Ti–O and Ti–O–Ti groups, respectively, and peaks at 982 and 604 cm^−1^ were assigned to Cu species [54]. The appearance of a peak at 604 cm^−1^ was only observed for V-Cu/Ti and V-Cu/BN-Ti, which was attributed to Cu(I)-O groups. This result was consistent with the XPS data, and it supported the notion that the formation of Cu^+^ species might contribute to oxidation of NH_3_.

The NH_3_-TPD and H_2_-TPR profiles were obtained in the temperature range 100–800 °C and the results were shown in Figure 4. Adsorption capacity (acid point, strength, and amount) for NH_3_ is one of the most critical factors in the SCR reaction [55,56]. NH_3_ adsorbed on the catalyst surface is converted to active-NH_3_ (i.e., absorbed-NH_3_ and NH_4_^+^) and then volatilized as nitrogen oxides. NH_3_-TPD was performed to study the adsorption of ammonia at the catalytic acid sites (Figure 4a). Peaks corresponding to the Brønsted acid sites of NH_3_ and NH_3_ species corresponding to V/Ti and Cu/Ti were confirmed at 434.6, 236.1, and 358.0 °C, respectively. For V-Cu/Ti and V-Cu/BN-Ti, two Brønsted acid sites were combined into a high-intensity peak compared to 269.1 and 316.7 °C for V-Cu/Ti, and 287.3 and 322.9 °C for V-Cu/BN-Ti. Table 2 lists the amount of acid sites in the synthesized catalysts derived from the NH_3_-TPD profiles. The amount of acid sites increased significantly in V-Cu/Ti and V-Cu/BN-Ti. This suggested that many NH_3_ species were formed at low temperatures, which means that they could be used in SCR and SCO of NH_3_ [57]. In the 500–800 °C range, peaks corresponding to NH_3_ and NH_3_ species adsorbed to Lewis acid sites were observed at 554.3, 565.4, 607.4, and 674.4 °C for V-Cu/BN-Ti, V-Cu/Ti, Cu/Ti, and V/Ti, respectively. NH_3_ adsorbed to Lewis acid sites participate in the reaction for the oxidation of NH_3_. In addition, a peak shift to a lower temperature and an increase in the amount of acid sites was confirmed. V-Cu/BN-Ti was identified as the optimal catalyst by confirming the increase of the Brønsted and Lewis acid sites and the peak shift to a lower temperature.

H_2_-TPR profiles (Figure 4b) confirm the reduction ability of the catalysts. The peak at 431.4 °C detected in V/Ti correspond to the reduction of Ti^4+^, which shifted to 339.0 °C in Cu/Ti [58]. Therefore, the copper species reduced Ti^4+^ at a lower temperature than the vanadium species. In Cu/Ti, the two peaks at low temperatures are related to the copper oxidation states transition (Cu^2+^ → Cu^+^ → Cu^0^), corresponding to highly dispersed CuO species (denoted as α) and CuO species strongly bound to the support (denoted as β) [59,60,61]. Upon the addition of vanadium, the intensity of the α-peak gradually increased, indicating that the combination of vanadium and copper led to the formation of more dispersed copper species. In Table 2, the hydrogen consumption calculated based on the TPR profiles showed that Cu^+^ was further reduced, which is consistent with the XPS data.

### 3.3. Catalytic Performance of NH_3_-SCR and NH_3_-SCO

The NH_3_/NO_X_ ratio in the SCR system is a critical factor affecting denitrification efficiency [62]. Considering that the ratio of NH_3_/NO_X_ should range from 0.8 to 1.2 for the effective removal of NOx in stationary, the selective oxidation reaction of ammonia in the NH_3_-SCR system of the synthesized catalysts was NH_3_/NO_X_. The evaluation of catalytic performance used a fixed-bed quartz reactor under the condition of NH_3_/NO_X_ = 1.0. Figure 5a shows the NH_3_-SCR performance of the catalysts measured at 20 °C intervals from 200 to 300 °C. V-Cu/BN-Ti exhibit improved NO_X_ conversion over V/Ti and Cu/Ti in the 200–260 °C interval (88% at 240 °C and 95% at 280 °C). This was ascribed to the high V^5+^/V^4+^ ratio and Cu^2+^ content. The increased surface exposure content by h-BN is also a contributing factor. NH_3_ might be oxidized in a side reaction to NO_X_ within the NH_3_ conversion (Figure 5b) and all the synthesized catalysts showed high N_2_ selectivity with an efficiency of >97%at all temperatures. However, the NO_X_ conversion of V-Cu/BN-Ti was equal or decreased compared with the other three catalysts at 260–300 °C. From Figure 5a,b, it seemed that high NH_3_ conversion of V-Cu/BN-Ti resulted in the absence of NH_3_ as a reductant for SCR reaction. Therefore, as the concentration of NH_3_ in flue gas decreased, the NH_3_/NO_X_ ratio changed and resulted in decreased NO_X_ conversion. Nevertheless, the reduced NO_X_ conversion still had an efficiency of over 90%, suggesting that the residual NO_X_ at the bottom of the catalyst layer can be successfully removed. V-Cu/Ti exhibit higher NO_X_ conversion and lower NH_3_ conversion than conventional V/Ti and Cu/Ti samples. From the TEM image of Appendix A, the copper species that contributes to the NH_3_ oxidation reaction was reduced by the competitive adsorption of vanadium and copper due to a large amount of TiO_2_ aggregation. NH_3_-SCO was calculated using Equation (11) to effectively compare selective oxidation to NH_3_ within the SCR system, as shown in Figure 6.

Evaluations performed under NH_3_/NO_X_ = 1:1 revealed that V/Ti showed excellent NH_3_-SCR performance over the entire temperature intervals (200–300 °C) but no oxidation of NH_3_. Cu/Ti exhibit the same NH_3_-SCR activity as V/Ti and showed oxidation of NH_3_. This was consistent with the characteristics of Cu^2+^ species from XPS analysis, resulting in lower NH_3_-Slip content. The lack of NH_3_-SCO activity of V-Cu/Ti is ascribed to the local competitive adsorption of vanadium and copper due to the aggregation of TiO_2,_ as shown in Figure 5a. V-Cu/BN-Ti showed the highest catalytic performances for both NH_3_-SCR and NH_3_-SCO (3% and 6% at 240 and 260 °C, respectively) among the four samples and only 9% and 2% NH_3_ was emitted at 240 and 260 °C, respectively. In fact, the oxidation performance for NH_3_ increased in all the sections compared with the single impregnated Cu/Ti. In summary, when copper was added to vanadium, the presence of Cu^+^ was able to selectively oxidize the remaining NH_3_ used in NH_3_-SCR. Furthermore, the use of h-BN negated the agglomeration of TiO_2_ as a support and enhanced the catalytic performance by increasing the surface exposure of the active species.

Figure 7 shows a comparison of the activities of V-Cu/Ti and V-Cu/BN-Ti, revealing high NH_3_-SCR performances in the presence of SO_2_. Generally, the presence of SO_2_ in exhaust gas reduces catalytic efficiency and impairs NH_3_ adsorption on the catalyst surface. In particular, the SCR catalyst impregnated with transition metal oxides was critical because of the presence of SO_2_ caused poisoning on the catalyst surface [31]. Figure 7a shows that the NO_X_ conversion performance decreased by approximately 20% for both catalysts over the entire temperature range due to the addition of SO_2_: V-Cu/Ti showed approximately 87.4% and V-Cu/BN-Ti 84.3% at 240 °C. However, the NH_3_ conversion of V-Cu/BN-Ti was the highest, 90% at 260 °C.

The performance of all the catalysts tended to decrease because NH_3_ was not used as a reducing agent in the NH_3_-SCR reaction in the presence of SO_2_, which caused a reduction in the proportion used for SCR and SCO. Nevertheless, V-Cu/BN-Ti increased the selective oxidation performance of NH_3_ (Figure 8) in all the sections, which resulted in enhanced oxidation at high NH_3_-Slip concentrations. Therefore, V-Cu/BN-Ti has the potential of reducing the emission of NH_3_-Slip by NH_3_-SCO in the SCR reaction in a SO_2_ atmosphere.

## 4. Conclusions

We synthesized a V-Cu/BN-Ti catalyst using the impregnation method to effectively oxidize NH_3_-Slip on the NH_3_-SCR. The crystal phases of TiO_2_ and h-BN were confirmed by XRD analysis. HR-TEM images confirmed that the dispersion of the CuO (001) lattice phase and TiO_2_ (101) particles were enhanced by using h-BN as support. The high V^5+^ ratio obtained by XPS analysis contributed to NH_3_-SCR catalytic efficiency, and the formation of Cu^+^ increased the oxidative performance of NH_3_. The formation of various oxidation states showed an increase in Brønsted and Lewis acid sites in the NH_3_-TPD and H_2_-TPR profiles and peak shift of acid sites to lower temperatures. Despite similar catalytic properties, V-Cu/Ti and V-Cu/BN-Ti differ in their catalytic efficiency for NH_3_-SCR and NH_3_-SCO. h-BN increased the surface exposure of active species. V-Cu/BN-Ti efficiently performed the NH_3_-SCR reaction even in an SO_2_ atmosphere and reduced the NH_3_-slip through the oxidation reaction of the remaining NH_3_. In conclusion, based on the results of the characterization and catalytic performance evaluation, V-Cu/BN-Ti showed 98% NO_X_ conversion, 98% N_2_ selectivity, and only 2% NH_3_-Slip at 260 °C. Therefore, it is a promising catalyst for the removal of residual NO_X_ and may be appropriate for ammonia emission regulation, even in the colder regions at the bottom SCR catalyst layers.

## Figures and Tables

**Figure 1 nanomaterials-12-02329-f001:**
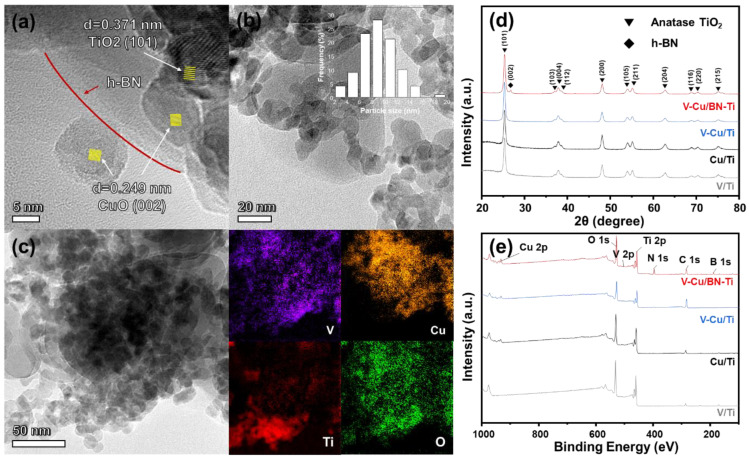
(**a**) High magnification TEM image, (**b**) TEM image and histogram of particle size distribution, (**c**) EDS mapping for V-Cu/BN-Ti, (**d**) XRD patterns, (**e**) XPS survey scan for V/Ti (gray line), Cu/Ti (black line), V-Cu/Ti (blue line), and V-Cu/BN-Ti (red line).

**Figure 2 nanomaterials-12-02329-f002:**
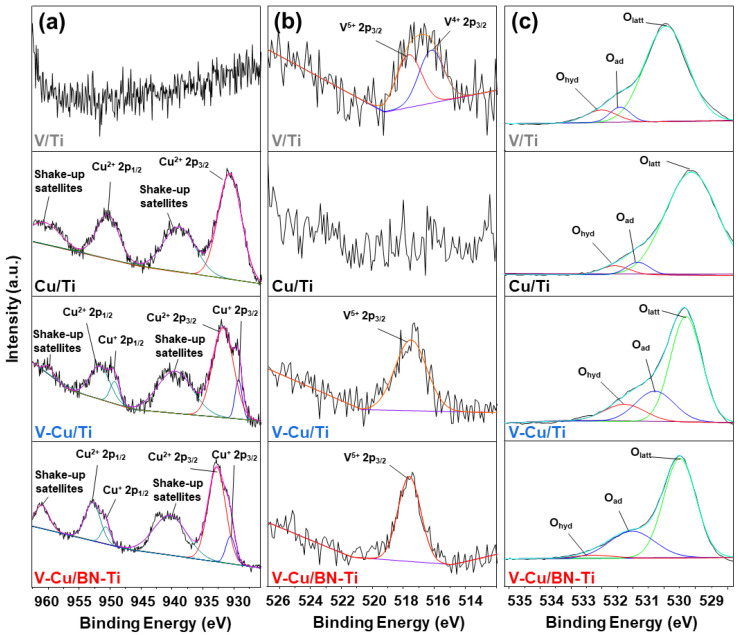
(**a**) Cu 2p, (**b**) V 2p, and (**c**) O 1s XPS spectra of the chemical states of the synthesized catalysts.

**Figure 3 nanomaterials-12-02329-f003:**
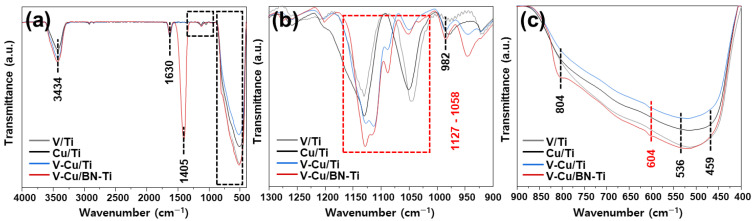
Acidity properties analysis of the synthesized catalysts. (**a**) Full FT-IR spectra and magnified, (**b**) 1300–900 cm^−1^, and (**c**) 900–400 cm^−1^ regions. V/Ti, gray line; Cu/Ti, black line; V-Cu/Ti, blue line; V-Cu/BN-Ti, red line.

**Figure 4 nanomaterials-12-02329-f004:**
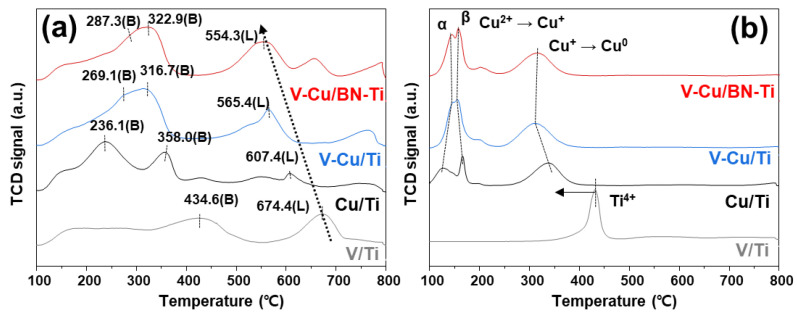
(**a**) NH_3_-TPD and (**b**) H_2_-TPR profiles for V/Ti (gray line), Cu/Ti (black line), V-Cu/Ti (blue line), and V-Cu/BN-Ti (red line).

**Figure 5 nanomaterials-12-02329-f005:**
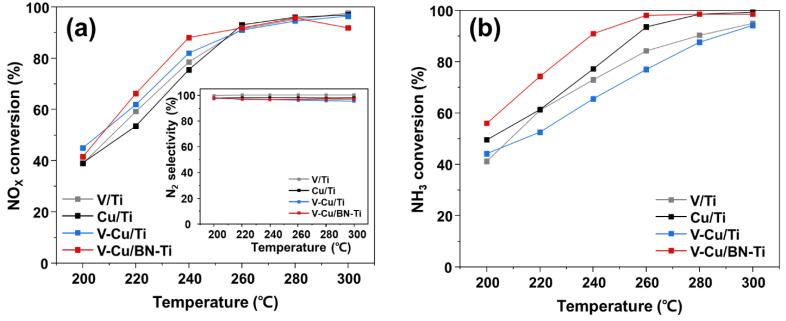
Catalytic performance of (**a**) NO_X_ conversion (Inset: N_2_ selectivity) and (**b**) NH_3_ conversion of the synthesized catalysts. Gas condition were [NO_X_] = [NH_3_] = 300 ppm, [O_2_] = 5 vol. %, [N_2_] = balance, total flow = 500 mL/min and GHSV = 60,000 h^−1^.

**Figure 6 nanomaterials-12-02329-f006:**
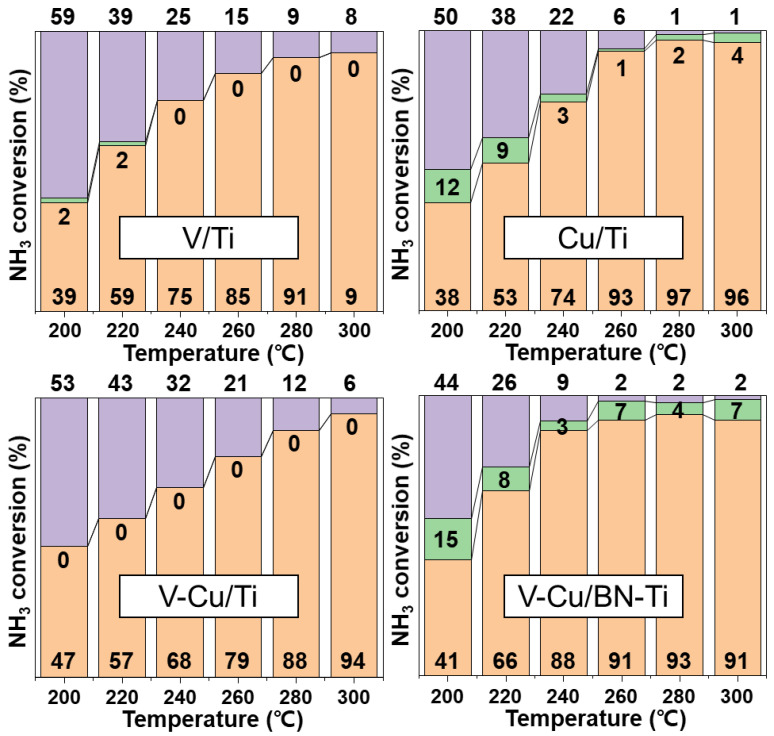
Slipped NH_3_ oxidation performance during NH_3_ conversion by the synthesized catalysts, calculated using Equation (11). NH_3_-SCR, orange bar; NH_3_-SCO, green bar; and NH_3_-Slip, purple bar. Gas conditions were [NO_X_] = [NH_3_] = 300 ppm, [O_2_] = 5 vol. %, [N_2_] = balance, total flow = 500 mL/min, and GHSV = 60,000 h^−1^.

**Figure 7 nanomaterials-12-02329-f007:**
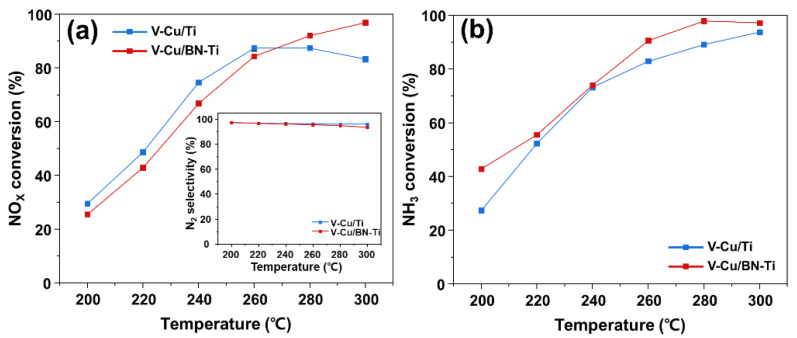
Catalytic performance with SO_2_ in (**a**) NO_X_ conversion, N_2_ selectivity, and (**b**) NH_3_ conversion of the synthesized catalysts. Gas conditions were [NO_X_] = [NH_3_] = 300 ppm, [SO_2_] = 100 ppm, [O_2_] = 5 vol. %, [N_2_] = balance, total flow = 500 mL/min, and gas hourly space velocity (GHSV) = 60,000 h^−1^.

**Figure 8 nanomaterials-12-02329-f008:**
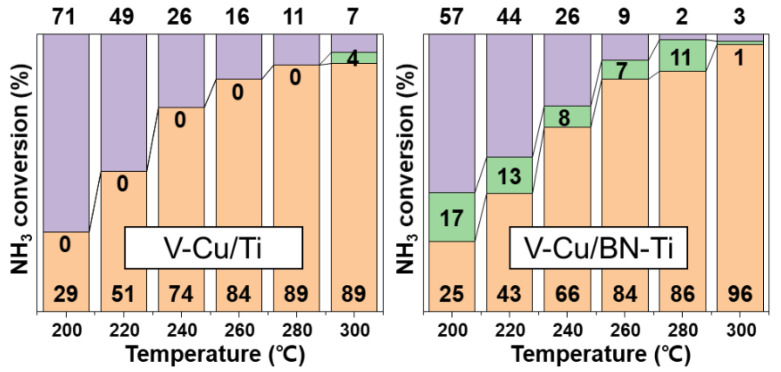
Slipped NH_3_ oxidation performance during NH_3_ conversion with SO_2_ of the synthesized catalysts, NH_3_-SCR (orange bar), NH_3_-SCO (green bar) and NH_3_-Slip (purple bar) calculated using Equation (11). Gas conditions were [NO_X_] = [NH_3_] = 300 ppm, [SO_2_] = 100 ppm, [O_2_] = 5 vol. %, [N_2_] = balance, total flow = 500 mL/min, and GHSV = 60,000 h^−1^.

**Table 1 nanomaterials-12-02329-t001:** Content of surface-exposed elements and valence states of elements and ratios of the synthesized catalysts.

Catalysts	Content of Surface-Exposed Elements (at %)		Composition of Copper Species (at %)			Composition of Oxygen Species (at %)
	V	Cu	Cu^2+^	Cu^+^	Cu^+^/(Cu^+^ + Cu^2+^)	O_α_/(O_α_ + O_β_)
V/Ti	0.84	-	-	-	-	6.53
Cu/Ti	-	3.01	100	0	0	5.20
V-Cu/Ti	0.77	2.96	66.39	8.67	0.11	22.58
V-Cu/BN-Ti	3.07	3.70	57.09	8.22	0.12	26.73

**Table 2 nanomaterials-12-02329-t002:** NH_3_ desorption and H_2_ consumption with regard to the acidity properties of the synthesized catalysts.

Catalysts	Brønsted Acid Sites (mmol/g)	Lewis Acid Sites (mmol/g)	H_2_ Consumption (μmol/g)
V/Ti	0.72	0.20	1.03
Cu/Ti	0.80	0.10	1.28
V-Cu/Ti	0.74	0.47	1.37
V-Cu/BN-Ti	0.88	0.45	1.46

## Data Availability

Not applicable.

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
