# Peer review of "High-Dispersed V2O5-CuOX Nanoparticles on h-BN in NH3-SCR and NH3-SCO Performance"

_nanomaterials, 2022, doi:10.3390/nano12142329_

Round 1

Reviewer 1 Report

In this manuscript, the authors synthesized a V-Cu/BN-Ti catalyst using the impregnation method to effectively oxidize NH3-Slip on the NH3-SCR. The formation of Cu+ at Cu 2p increased the oxidative performance of NH3. The formation of various oxidation states showed an increase in Brønsted and Lewis acid sites, which affected the catalytic performance. Generally, this work is very well performed and the data are of high quality, moreover, the findings in this manuscript can also provide insights into the thermal catalyst design. For all these reasons I recommend publication of this paper after some revisions:

(1) Did the authors take into account the effects of internal and external diffusion on the catalytic performance during the catalytic reaction? Whether internal diffusion and external diffusion are eliminated.

(2) For V-Cu/Ti and V-Cu/BN-Ti catalysts, what are the contents of V and Cu, respectively? Different contents of V and Cu also lead to different catalytic performance.

(3) In Figure 5a, compared with the other three catalysts, under the reaction of 280-300 oC, the NOx conversion rate of the V-Cu/BN-Ti catalyst is reduced. What is the reason for this? The author needs to give reasonable explanation

(4) Will the particle size of the active components V and Cu species in the catalyst grow after the catalytic reaction is over? Authors are advised to provide TEM images for verification.

(5) During the catalytic reaction process, the morphology of the catalyst plays an important role for the catalytic activity. Currently, some special morphologies, such as hollow multi-shelled structure and hierarchical porous structure are extensively used in catalysis, and the following papers are encouraged to be cited: Adv. Funct. Mater. 2019, 29, 1806588. Catal. Commun. 2019, 129, 105729. Mater. Chem. Front., 2021, 5, 1126-1139.

Author Response

We express the gratitude to the reviewer for your suggestions and insights. The responses to all comments have been prepared and attached. Please see the attachment.

Reviewer 2 Report

This paper describes the preparation of catalysts for NH3-SCR processes. These catalysts are synthesized by the immobilization of transition metals V, Cu and mixtures of both on TiO2. In addition to increasing the surface area and hence the catalyst’s contact area of the catalyst’s boron nitride has been used in the synthetic procedure. It is an interesting work with interesting results, but the manuscript needs to be improved to attain the standards. Several English mistakes are found throughout the text, but what is more important the draft is difficult to read. It must be corrected by an English speaker before being considered for publication.

For instance:

Researches has been devoted toward been done on the application…

Can be more formed for low…

To observe exist of

It found that peaks were appeared..

And adsorption on the catalyst surface was reduced the possibility…

What are textiles properties?

And so on…

The Catalyst preparation section must be improved. TiO2 does not render solutions but suspensions. The same with boron nitride…

What is the metallic precursor used to immobilize copper(II)

Page 6, line 29 h-BN improves particle agglomeration… are you sure? Or maybe avoids particle agglomeration

Figure 3c in the text should be Figure 2c

Page 8 line 15, this corresponds to C-OH stretching and OH bending… how do you explain the presence of C-OH stretching bands?

Page 8, line 324 What are the expected “new functional groups”?

Page 10 line 377, In understand the temperature range is from 200ºC and not 20oC

Author Response

(The authors gave the same response as above.)

Round 2

Reviewer 1 Report

The authors have carefully revised the manuscript and made satisfactory response to me. I think the revised manuscript is suitable for publication in Nanomaterials at this state. However, the subscripts of some chemical formulas in the references need to be modified.

Author Response

The responses to all comments have been prepared and attachen upload file. thank you.

Reviewer 2 Report

Although many of the suggested corrections explicitily mentioned have been performed by the authors, extensive editing of English language and style are still missing. 

In addition, I got a critical comment, the copper metallic precursor copper (II) nitrate has been included as CuN2O6.3H2O... what type of formulation is this according to IUPAC rules?

I really think that our role as researchers is to serve as models for students and young researchers. If we do not take care o this type of detail we are failing. Nomenclature and formulation are the languages we use to communicate in chemistry, so please take care...

Author Response

(The authors gave the same response as above.)
